# Effect of stimulus orientation and intensity on short-interval intracortical inhibition (SICI) and facilitation (SICF): A multi-channel transcranial magnetic stimulation study

Sergei Tugin[1,2⊙]*, Victor H. Souza[1,2,3⊙], Maria A. Nazarova[4,5], Pavel A. Novikov[4], Aino E. Tervo[1,2], Jaakko O. Nieminen[1,2], Pantelis Lioumis[1,2], Ulf Ziemann[6,7], Vadim V. Nikulin[4,8], Risto J. Ilmoniemi[1,2]

1 Department of Neuroscience and Biomedical Engineering, Aalto University School of Science, Espoo, Finland, 2 BioMag Laboratory, University of Helsinki and Helsinki University Hospital, HUS Medical Imaging Centre, Helsinki, Finland, 3 School of Physiotherapy, Federal University of Juiz de Fora, Juiz de Fora, MG, Brazil, 4 Centre for Cognition and Decision Making, Institute for Cognitive Neuroscience, National Research University Higher School of Economics, Moscow, Russia, 5 Federal State Budgetary Institution "Federal Center of Brain Research and Neurotechnologies" of the Federal Medical Biological Agency, Moscow, Russia, 6 Department of Neurology and Stroke, Eberhard Karls University, Tübingen, Germany, 7 Hertie Institute for Clinical Brain Research, Eberhard Karls University, Tübingen, Germany, 8 Department of Neurology, Max Planck Institute for Human Cognitive and Brain Sciences, Leipzig, Germany

⊙ These authors contributed equally to this work.
* sergei.tugin@aalto.fi

**Data Availability Statement:** According to our ethical permission statement, we cannot make

## Abstract

Besides stimulus intensities and interstimulus intervals (ISI), the electric field (E-field) orientation is known to affect both short-interval intracortical inhibition (SICI) and facilitation (SICF) in paired-pulse transcranial magnetic stimulation (TMS). However, it has yet to be established how distinct orientations of the conditioning (CS) and test stimuli (TS) affect the SICI and SICF generation. With the use of a multi-channel TMS transducer that provides electronic control of the stimulus orientation and intensity, we aimed to investigate how changes in the CS and TS orientation affect the strength of SICI and SICF. We hypothesized that the CS orientation would play a major role for SICF than for SICI, whereas the CS intensity would be more critical for SICI than for SICF. In eight healthy subjects, we tested two ISIs (1.5 and 2.7 ms), two CS and TS orientations (anteromedial (AM) and posteromedial (PM)), and four CS intensities (50, 70, 90, and 110% of the resting motor threshold (RMT)). The TS intensity was fixed at 110% RMT. The intensities were adjusted to the corresponding RMT in the AM and PM orientations. SICI and SICF were observed in all tested CS and TS orientations. SICI depended on the CS intensity in a U-shaped manner in any combination of the CS and TS orientations. With 70% and 90% RMT CS intensities, stronger PM-oriented CS induced stronger inhibition than weaker AM-oriented CS. Similar SICF was observed for any CS orientation. Neither SICI nor SICF depended on the TS orientation. We demonstrated that SICI and SICF could be elicited by the CS perpendicular to the TS, which indicates that these stimuli affected either overlapping or strongly connected neuronal populations. We concluded that SICI is primarily sensitive to the CS intensity and that CS intensity adjustment resulted in similar SICF for different CS orientations.

physiological or anatomical data publicly available. However, the data can be accessed upon reasonable request if the confidentiality requirements are strictly followed. The data can be requested from researchdata@aalto.fi.

**Funding:** This project has received funding from the European Research Council (ERC) under the European Union's Horizon 2020 research and innovation programme (grant agreement No 810377), the Jane and Aatos Erkko Foundation, the Academy of Finland (Decisions No. 294625, 306845, and 327326) and the Finnish Cultural Foundation. Pavel A. Novikov was funded by Aalto AScI Visiting Researcher Programme and by RFBR, project number 20-315-70048. Maria A. Nazarova was funded by RFBR, project number 20-315-70048.

**Competing interests:** Risto J. Ilmoniemi is an advisor and a minority shareholder of Nexstim Plc. Jaakko O. Nieminen and Risto J. Ilmoniemi are inventors on patents and patent applications on multi-channel TMS technology. Ulf Ziemann has received grants from Bristol Myers Squibb, Janssen Pharmaceutica NV, Takeda, and personal fees from Bayer Vital GmbH, Pfizer GmbH, CorTec GmbH, all not related to this work. Pantelis Lioumis has served as consultant to Nexstim Plc. for purposes other than this study. The other authors declare no conflict of interest.

**Abbreviations:** AL, anterolateral; AM, anteromedial; APB, abductor pollicis brevis; CS, conditioning stimulus; CSI, conditioning stimulus intensity; CSO, conditioning stimulus orientation; E-field, electric field; EMG, electromyography; GABA$_A$, gamma-aminobutyric acid A; ICF, intracortical facilitation; IQR, interquartile range; ISI, interstimulus interval; I-wave, indirect wave; MEP, motor evoked potential; mTMS, multi-channel TMS; PL, posterolateral; PM, posteromedial; RMT, resting motor threshold; SICF, short-interval ICF; SICI, short-interval intracortical inhibition; TMS, transcranial magnetic stimulation; TS, test stimulus; TSO, TS orientation.

# Introduction

Manual dexterity, along with motor learning, requires fine-tuning of neuronal activity in the motor cortex, which is mediated through discrete interactions of excitatory and inhibitory interneuronal circuits [1]. Unbalanced inhibitory and excitatory processes disrupt the execution of fine movements, as observed, for example, in musicians' dystonia [2,3]. Paired-pulse transcranial magnetic stimulation (TMS) is widely used to assess the intracortical inhibitory and excitatory processes, offering a possibility to probe non-invasively the interactions of the cortical neuronal circuits in healthy and pathological conditions [4,5]. Fine adjustment of the stimulus intensities and interstimulus intervals (ISI) enables the assessment of both inhibitory and excitatory circuits. Despite extensive research on the effect of these parameters on motor evoked potentials (MEP), little is known about the dependency of inhibitory and facilitatory phenomena on the orientation of the TMS-induced electric field (E-field) in the cortex.

In conventional paired-pulse TMS, two consecutive stimuli induce an E-field in the same location in the cortex with the same E-field orientation. The stimuli follow each other with a millisecond-level ISI and interact so that the first, conditioning stimulus (CS), activates neurons that influence the MEP elicited by the second, test stimulus (TS) [6]. A sub-threshold CS 0.5–5 ms before a supra-threshold TS suppresses the MEP amplitude; this phenomenon is called short-interval intracortical inhibition or SICI [7,8]. However, once the ISI is in the range of 6–30 ms, the MEP amplitudes increase in what is called intracortical facilitation (ICF) [9]. Furthermore, a supra-threshold CS 1.3–1.7 or 2.3–3.0 ms before the TS enhances the MEP amplitudes [10], causing short-interval ICF (SICF). These three phenomena seem to have different neuronal origins. SICI at a 0.5–1-ms ISI may reflect neuronal refractoriness [11,12], and with an ISI of 1–5 ms, it is associated with postsynaptic inhibition mediated by gamma-aminobutyric acid A (GABA$_A$) interneurons. In contrast, SICF most likely has a non-synaptic origin through direct excitation of excitatory interneurons that mediate indirect (I-) waves [13]. In turn, ICF at longer ISIs is mediated by cortical neuronal circuits distinct from the circuits involved in the generation of SICI and SICF [8,14,15].

It is well known that the stimulus intensity and ISI are critical for generating SICI, SICF, and ICF [7,16,17]. Additionally, the orientation of the E-field induced in the cortex plays an important role in evaluating the neuronal excitatory and inhibitory circuits [8,18–20]. SICI, SICF, and ICF have mainly been studied with the E-field oriented approximately perpendicular to the central sulcus for both the CS and TS. Due to the need for a millisecond-level ISI between the CS and TS, a standard TMS coil allows the delivery of two consecutive pulses in the same or oppositely directed orientations. The experiments with oppositely directed orientations of the E-field plays an important role for the identification of the interneurons for generation of I-waves and SICF [21,22]. However, the differently oriented current has, in some studies, limited the minimal ISI to 3 ms [22,23] and not allowed to perform stimulation in other orientations. To allow stimulating in perpendicular orientations, Ziemann et al. placed two figure-of-eight coils on top of each other and demonstrated that ICF, but not SICI, was affected by a 90° clockwise rotation of the CS from the anteromedial (AM) to the posteromedial (PM) E-field orientation [8].

Recently, we assessed SICI and ICF was conducted by our group with a multi-channel TMS (mTMS) transducer enabling electronic rotation of the induced E-field [20]. This experiment demonstrated that SICI at 0.5-ms ISI and ICF were the strongest when both CS and TS were delivered in the same orientation, i.e., AM−AM or posterolateral−posterolateral (PL−PL), and were weaker when the CS was perpendicular to the TS. Thus, it was concluded that excitatory and inhibitory neuronal mechanisms exhibit a critical sensitivity to the stimulus orientation. The observations of Souza et al. [20] and Ziemann et al. [8] were based on the CS and TS

intensities set relative to the resting motor threshold (RMT) only in the AM orientation and not in the PM orientation, in which the RMT can be about 30% higher than in the AM orientation [24]. Therefore, it remains unclear whether the neuronal mechanisms engaged in the inhibitory and facilitatory phenomena are truly affected by the stimulus orientation rather than a decrease in the relative stimulation intensity. This experimental gap can be addressed by adjusting the CS intensity to the corresponding RMT in each orientation.

In this study, we used our mTMS system [25] to elaborate on the earlier findings [8,20] and assess how the mechanisms underlying SICI and SICF respond to changes in the CS and TS intensity and orientation. We tested AM and PM E-field orientations for both the CS and TS, with the stimulus intensities defined in terms of the RMT in the corresponding orientation. We also tested four stimulus intensities and two ISIs to generate SICI and SICF. Since SICI at 1–5 ms ISI requires activation of the inhibitory interneurons, it can be assumed that this SICI is insensitive to the E-field orientation. On the other hand, SICF requires preactivation of the pyramidal neurons and therefore the E-field orientation should be critical for it. Based on the above and also on previous studies [8,20], we hypothesized that a CS perpendicular to the TS would produce significant SICI and SICF. Additionally, changes in the CS orientation would critically impact SICF, while SICI phenomena would be more affected by changes in the CS intensity.

## Methods

### Subjects

Eight healthy volunteers (aged 21–35 years, five men) with no contraindications to TMS [26,27] participated in the study after giving written informed consent. The study was carried out in accordance with the Declaration of Helsinki and approved by the Coordinating Ethics Committee of the Hospital District of Helsinki and Uusimaa (number HUS/1198/2016).

### EMG recordings

Surface electromyography (EMG) was recorded from the right abductor pollicis brevis (APB) muscle. The skin was prepared with sandpaper and sanitized with 80% alcohol. The radiolucent electrodes (Ambu BlueSensor BR, Ambu A/S, Denmark) were placed in a belly–tendon montage with one electrode over the muscle belly and the other over the closest bony eminence. The ground electrode was placed on the dorsum of the right hand in the middle of the third metacarpal bone. EMG signals were digitized at 3 kHz, with a 10–500-Hz band-pass filter, using an eXimia EMG 3.2 system (Nexstim Plc, Finland).

### mTMS and stimulus definition

TMS was delivered using our custom-made mTMS system [25] including a transducer composed of two overlapping perpendicular figure-of-eight coils [28]. This system enables fast (sub-millisecond) and accurate (~1°) electronic control of the induced E-field orientation without mechanical movement of the transducer. The E-field strength in any orientation could be adjusted from 0 to 170 V/m. These intensity values refer to the stimulation at 15-mm depth in a spherically symmetric head model with the transducer bottom at 85-mm distance from the origin [29]. The E-field focality was similar for all orientations [28].

Single pulses had a monophasic pulse waveform with 60.0-, 30.0-, and 43.2-μs rising, holding, and falling phases, respectively [30]. The phase durations of the paired pulses differed based on the CS intensity, as specified in Table 1. The stimulus intensity in paired-pulse mTMS was adjusted by varying the rising-phase duration of each pulse [31,32], following the

**Table 1. TMS pulse durations of the conditioning (CS) and test stimuli (TS) at each CS intensity as a percentage of the resting motor threshold (RMT).**

| CS intensity (% RMT) | CS waveform durations: rising, holding, falling (µs) | TS waveform durations rising, holding, falling (µs) |
|---|---|---|
| 50 | 24.8, 30.0, 19.3 | 72.3, 30.0, 51.0 |
| 70 | 37.1, 30.0, 27.9 | 73.2, 30.0, 52.0 |
| 90 | 51.2, 30.0, 37.4 | 76.7, 30.0, 53.7 |
| 110 | 70.5, 30.0, 49.8 | 82.8, 30.0, 57.5 |

The values represent the duration of three phases of a trapezoidal monophasic waveform: Rising, holding, and falling.

procedure described in [33]. The E-field orientation across the central sulcus will be referred to as the AM orientation. The orientation where the E-field is rotated by 90˚ clockwise from the AM orientation, along the central sulcus, will be referred to as the PM orientation (Fig 1A). The AM and PM orientations were chosen due to the fact that they are widely applied in paired-pulse stimulation, albeit separately, i.e. in AM–AM or PM–PM combinations [34]. However, only a few studies have tested the combination of these orientations as a CS and TS [8,20].

## Experimental protocol

Subjects sat in a reclining chair with their right hand on a pillow in a pronated position. Structural T1-weighted magnetic resonance images (voxel size 1×1×1 mm$^3$) of each subject were obtained prior to the experiments. The mTMS transducer position relative to the subject's brain was monitored with the Nexstim NBS 3.2 neuronavigation system (Nexstim Plc). The mTMS transducer was placed tangential to the scalp over the left primary motor cortex. First, the APB muscle hotspot was determined as the stimulation site evoking the largest MEPs for a fixed suprathreshold stimulation intensity. Then, the optimal stimulation orientation at the APB hotspot was defined by delivering approximately 20 single pulses guided by an automatic algorithm based on Bayesian optimization [35]. Based on this optimal orientation, the mTMS

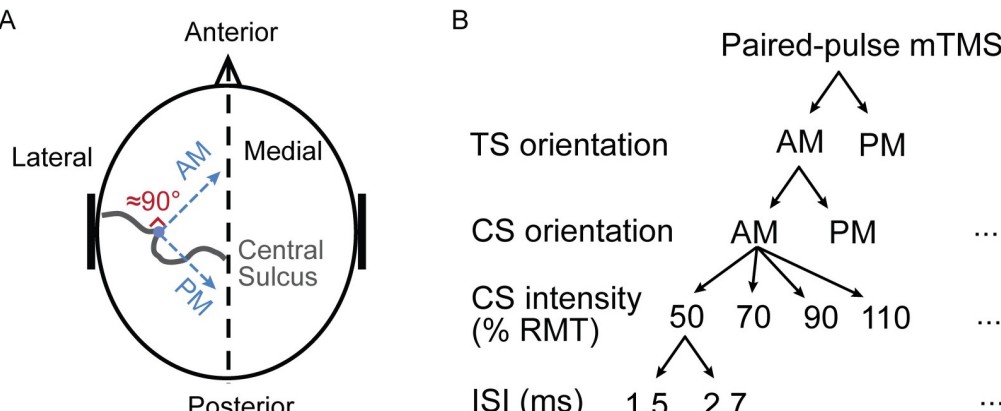

**Fig 1. Experimental protocol.** (A) Schematic representation of the stimulus orientations. The APB hotspot is indicated as a blue dot. The AM orientation of the E-field is approximately perpendicular to the central sulcus (gray curve). The PM is oriented 90˚ clockwise from the AM. (B) Schematic representation of paired-pulse stimulation. Thirty-two paired-pulse configurations were utilized based on the following features: Two orientations of the CS: AM and PM; two orientations of the TS: AM and PM; four CS intensities: 50, 70, 90, and 110% RMT; two ISIs: 1.5 and 2.7 ms. The ellipsis indicates an intentional omission of the subdivision for all configurations, except for the first one.

transducer was physically rotated by 90˚ clockwise, so that the top coil induced an E-field along the optimal orientation across the central sulcus. As a result, the bottom and top coils induced E-fields approximately across (AM orientation) and along (PM orientation) the central sulcus, respectively. The physical transducer rotation maximized the range of stimulus intensities in the PM orientation. Finally, we estimated the RMT in the AM and PM orientations by delivering 20 pulses with varying intensities according to a maximum-likelihood method [36].

## SICI and SICF protocols

We investigated the strength of SICI and SICF by varying the CS and TS orientations, CS intensities, and ISIs in paired-pulse mTMS (Fig 1B). The E-field for both the CS and TS was either AM- or PM-oriented, making four combinations in total: AM–AM, AM–PM, PM–PM, PM–AM, where the first and second terms refer to the CS and TS orientations, respectively. The CS intensity was 50, 70, 90, or 110% RMT, to evaluate its effects on SICI and SICF; the TS intensity was fixed to 110% RMT. The RMT refers to a value measured in the same stimulus orientation as the corresponding CS or TS. ISI was either 1.5 or 2.7 ms to maximize the sensitivity to SICF [7,37,38]. Additionally, SICI induced with approximately 1 ms and 2.5–3 ms ISI has physiologically distinct phases and these intervals are widely applied to induce inhibition in paired-pulse protocols [11,39–41]. Therefore, in total, 32 paired-pulse configurations were assessed. In addition, we recorded 20 reference MEPs due to single-pulse TMS with 110% RMT in the AM and PM orientations.

Pseudo-random intervals between consecutive pulse configurations (either paired or single pulses) were sampled from a uniform distribution ranging from 2.4 to 3.6 s. Each stimulus configuration was administered 20 times. The order of the pulses was pseudo-randomized, and the pulses were divided into ten blocks, followed by short breaks of about 2 min. At the beginning of each block, we administered an additional single-pulse stimulus in the AM and PM orientations (order pseudo-randomized for each sequence) to avoid an arousal effect after a break. MEPs evoked by these additional pulses were excluded from the analysis. In total, each subject received 700 pulses.

## Data analysis

MEPs were analyzed with a custom-made script written in Matlab 2018 (The MathWorks, Inc., USA). EMG data were divided into epochs from −1000 (pre) to 1000 ms (post) relative to the TMS pulse. The epochs were visually inspected; less than one percent of the epochs having muscle pre-activation greater than ±15 μV or movement artifacts were removed from the analysis. The MEP amplitude was automatically extracted as the peak-to-peak value within 15–60 ms after a TMS pulse. The effects of the CS orientation and intensity, ISI, and TS orientation on the MEP amplitudes were analyzed using a linear mixed-effects model [42]. Heteroscedasticity and deviation from normality in MEP amplitude, i.e., unequal variance across the configurations and positive skewness in the data distribution, were corrected using a log transformation before modeling. Each configuration and all possible interactions were modeled as fixed effects. Subject identifiers were modeled as a random effect with correlated random intercepts and slopes for the CS and TS. The model parameters and diagnostics are similar to the model described in [20]. *F*- and *p*-values for the selected model were computed with a Type-III analysis of variance using Satterthwaite's method. Estimated marginal means and *p*-values were corrected for false discovery rate and applied for post-hoc multiple comparisons. Modeling was performed in R 3.6 (R Core Team, Austria) language using the *lmer* and *afex* packages. All other statistical analysis was performed in Matlab R2020a.

To quantify the level of SICI and SICF for each subject, we calculated the ratio between the median MEP amplitude in each paired-pulse configuration and the median single-pulse MEP amplitude in the corresponding TS orientation. Two-sample t-tests were performed to assess inhibitory and facilitatory effects relative to a single pulse in each orientation. The median was preferred over the mean to avoid a bias from the outliers, which accounted for 2.8% of all amplitude samples. MEP amplitudes that exceeded 1.5 times the interquartile range (IQR) were considered as outliers. The outliers were calculated for each configuration and for each subject to support the choice between median or mean, and they were not removed from the analysis. To compare the effects induced by the paired pulses (inhibition or facilitation), we divided a higher value of amplitude induced by one configuration by a lower value of amplitude induced by the other configuration; thereby, the results are presented as ratios between higher and lower amplitudes. Spearman's linear correlation coefficient was computed to identify a possible systematic change in the single-pulse MEP amplitude throughout the experiment. Two-sample t-tests were used to compare the RMTs in the PM and AM orientations. The threshold for statistical significance was set at $p = 0.05$.

## Results

The median E-field strength at the RMT was 33% higher in the PM orientation than in the AM orientation (PM: 112 V/m, IQR: 99–120 V/m; AM: 85 V/m, IQR: 67–91 V/m) ($p < 0.01$). Single-pulse TMS with 110% RMT resulted in similar MEP amplitudes for the AM (median: 640 μV, IQR: 357–1789 μV) and PM (median 421 μV, IQR: 113–931 μV) orientations ($p = 0.48$). The single-pulse MEP amplitude across subjects was stable during the experiment, indicating that no evident MEP habituation occurred ($p = 0.27$ and $p = 0.28$ for the single pulses in the AM and PM orientations, respectively) [43,44].

### Dependence of SICI on stimulation parameters

Fig 2 shows the MEP amplitudes resulting from the linear mixed-effects model for all tested paired-pulse mTMS configurations; the statistical model results are depicted in Table 2. The significant differences between MEP responses to paired-pulse stimulation are presented in *S1–S4 Tables*. The MEP amplitude was significantly affected by the interaction between the CS orientation, ISI, and CS intensity. Namely, at 1.5-ms ISI, when the TS was delivered in the AM orientation, with 50% RMT CS intensity, the AM-oriented CS led to 1.5 times stronger MEP inhibition than in the PM-oriented CS. In contrast, when the TS was either in the AM or PM orientation, with 70% RMT CS intensity and at 2.7-ms ISI, the inhibition for the PM-oriented CS was at least 1.6 times as strong as the inhibition for the AM-oriented CS. Additionally, at both ISIs, when the TS was in the AM orientation with 90% RMT CS intensity, we observed an approximately 1.5 times stronger inhibition for the PM-oriented CS than for the AM-oriented CS.

With 70% RMT CS intensity, inhibition was observed in all orientations of the CS and TS at 2.7-ms ISI. Additionally, with the same CS intensity, we detected inhibition at 1.5-ms ISI when the TS was AM-oriented. Compared with 50% RMT CS intensity, 70% RMT CS intensity induced a stronger inhibitory effect only at 2.7-ms ISI; compared with 90% RMT CS intensity, it led to stronger inhibition at both 1.5- and 2.7-ms ISIs.

Stimuli delivered at 2.7-ms ISI induced inhibition with 90% RMT CS intensity in all CS and TS orientations. Compared with stimulation at 1.5-ms ISI, paired-pulse mTMS at 2.7-ms ISI induced stronger SICI in all CS and TS orientations with 70 and 90% RMT CS intensities. At 1.5-ms ISI, neither inhibition nor facilitation was produced with 90% RMT CS intensity.

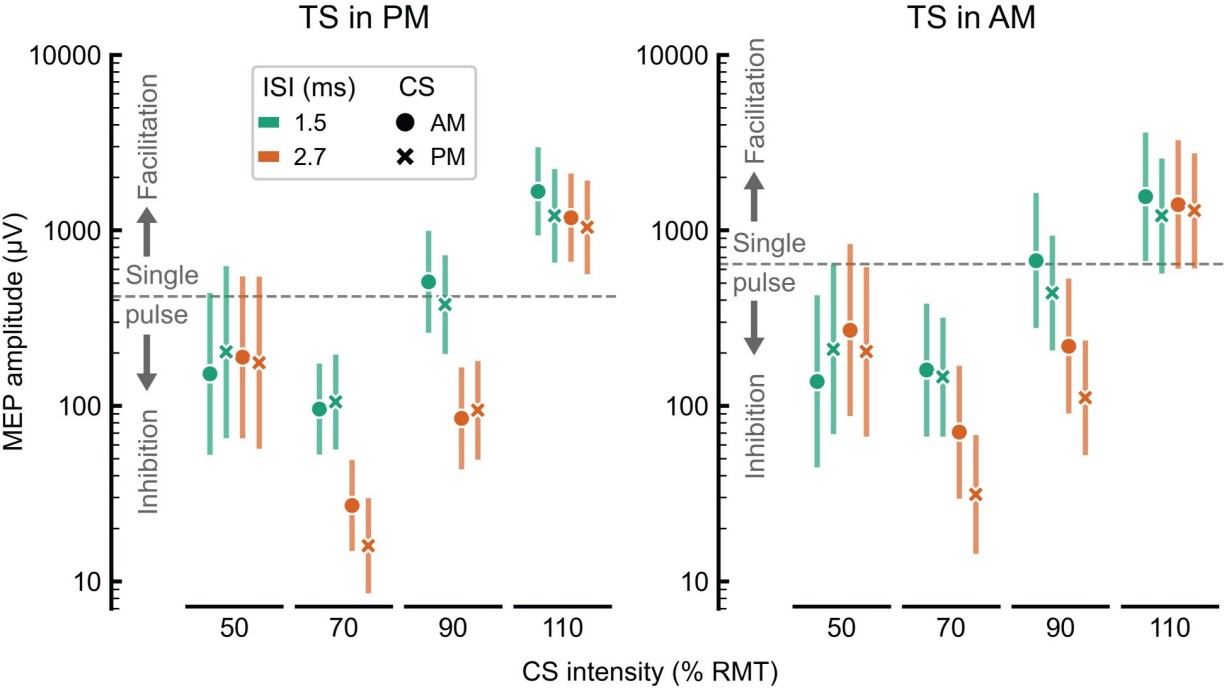

**Fig 2. MEP amplitude from the linear mixed-effects model as a function of the CS and TS orientation and the CS intensity at 1.5- and 2.7-ms ISIs.** The median single-pulse MEP amplitude is marked by the dashed horizontal line. The left and right panels represent the data for the TS in the AM and PM orientations, respectively. MEPs induced by paired-pulse stimulation are grouped by the CS intensities (50, 70, 90, and 110% RMT), and subdivided into two ISIs (1.5 ms–green, and 2.7 –orange), and CS orientations (AM orientation, dots; PM orientation, crosses). The error bars represent the 95% confidence interval of the estimated marginal means.

### Dependence of SICF on stimulation parameters

Supra-threshold CS intensity (110% RMT) generated SICF at all ISIs and CS and TS orientations. Both CS orientations induced a similar amount of SICF at 1.5- and 2.7-ms ISIs in all cases, with an exception for SICF at the AM–PM-oriented stimuli (at 1.5-ms ISI, the median relative MEP amplitude was 1.4 times higher than the median relative MEP amplitude at 2.7-ms ISI; $p$ = 0.03).

## Discussion

Using paired-pulse TMS with the intensities normalized to the RMT in different orientations, we demonstrated that the orientation-specific CS intensity, but not the E-field orientation affects the level of SICI. Stronger SICI was induced by the CS in the PM orientation compared to the conventional AM orientation, with a CS intensity of 70 or 90% RMT, due to stronger CS intensities in the PM orientation compared to the intensities in the AM. We also revealed that SICI and SICF can be induced by any CS–TS combination in both AM and PM orientations with suitable CS intensity and ISI. We found that SICI follows a U-shaped dependency on the CS intensity at both 1.5- and 2.7-ms ISIs and in any orientation of the CS and TS, as has been previously reported only for the AM-oriented stimuli [10]. In turn, differently oriented CS and TS at 110% of the orientation-specific RMT induced similar SICF across almost all tested configurations.

### Effect of stimulus parameters on inhibitory circuits

The CS in the PM orientation produced stronger inhibition than in the AM orientation. A likely explanation is that a 33% higher stimulus intensity in the PM orientation might have

**Table 2. Type-III analysis of variance table for the linear mixed-effects model analysis of the MEP amplitude.**

| Effect | DoF (numerator, denominator) | *F*-value | *p*-value |
|---|---|---|---|
| TSO | (1, 7.0) | 1.91 | 0.209 |
| CSO | (1, 7.1) | 3.75 | 0.093 |
| ISI | (1, 5048.0) | 361.58 | **< 0.001** |
| CSI | (3, 7.0) | 113.04 | **< 0.001** |
| TSO × CSO | (1, 5048.0) | 5.4 | **0.02** |
| TSO × ISI | (1, 5048.0) | 16.68 | **< 0.001** |
| CSO × ISI | (1, 5048.0) | 10.7 | **0.001** |
| TSO × CSI | (3, 5048.0) | 12.29 | **< 0.001** |
| CSO × CSI | (3, 5048.0) | 7.41 | **< 0.001** |
| ISI × CSI | (3, 5048.0) | 131.85 | **< 0.001** |
| TSO × CSO × ISI | (1, 5048.0) | 3.83 | 0.05 |
| TSO × CSO × CSI | (3, 5048.0) | 2.48 | 0.059 |
| TSO × ISI × CSI | (3, 5048.0) | 0.25 | 0.858 |
| CSO × ISI × CSI | (3, 5048.0) | 8.77 | **< 0.001** |
| TSO × CSO × ISI × CSI | (3, 5048.0) | 0.98 | 0.402 |

The fixed factors include the test stimulus orientation (TSO), conditioning stimulus orientation (CSO), interstimulus interval (ISI), and conditioning stimulus intensity (CSI). The interaction between factors is represented as "×". The table contains the numerator and denominator degrees of freedom (DoF) for each effect, followed by the *F*- and *p*-values. The values were computed with Satterthwaite's method. *p*-values in bold are smaller than the threshold for statistical significance of 0.05.

resulted in a more effective activation of GABAergic interneurons, which, in turn, led to stronger SICI. These interneurons have an isotropic dendritic arborization with no preferential somatodendritic orientation [45,46] and, therefore, likely have no clear preferential direction of activation [8,20]. Thereby, a higher intensity might activate them more effectively or recruit an additional population of inhibitory interneurons.

The hypothesis that stimulus intensities, but not the orientations, determine the SICI level is supported by previous studies [8,20]. In particular, Ziemann et al. [8] reported that at 1−3-ms ISI, the CS in the AM and PM orientations induced similar levels of SICI for an AM-oriented TS. Moreover, in our previous study with the same stimulation intensity in four CS orientations (AM, PM, PL, and anterolateral (AL)), we observed no significant changes in SICI at 2.5-ms ISI, with the only exception that the PL orientation showed stronger SICI compared with the PM and AL orientations [8,20]. It is important to note that both of those applied only one CS intensity, either 5% of the stimulator output below active motor threshold [8] or 80% RMT [20]. These intensities typically result in maximum SICI [10], with the possible consequence of a floor effect, resulting in low sensitivity of detecting SICI changes with a variation of the CS orientation.

Additionally, the notion that the CS intensity rather than the orientation is critical for SICI is supported by the studies where the CS orientation was opposite to the AM, namely the PL [34,47,48]. In these studies, the RMT in the PL orientation was approximately 30% higher than in the AM orientation, which is comparable with the intensity difference between the PM vs. AM orientations in the present study. Stronger SICI was detected in the PL−PL orientation compared with the AM−AM stimulation when stimulus intensity was adjusted to the RMT in the corresponding orientation [34,47,48].

Finally, the individual data in Fig 3 show that the SICI intensity curves for the CS in the AM vs. PM orientations are similar intraindividually for most subjects, but the SICI intensity

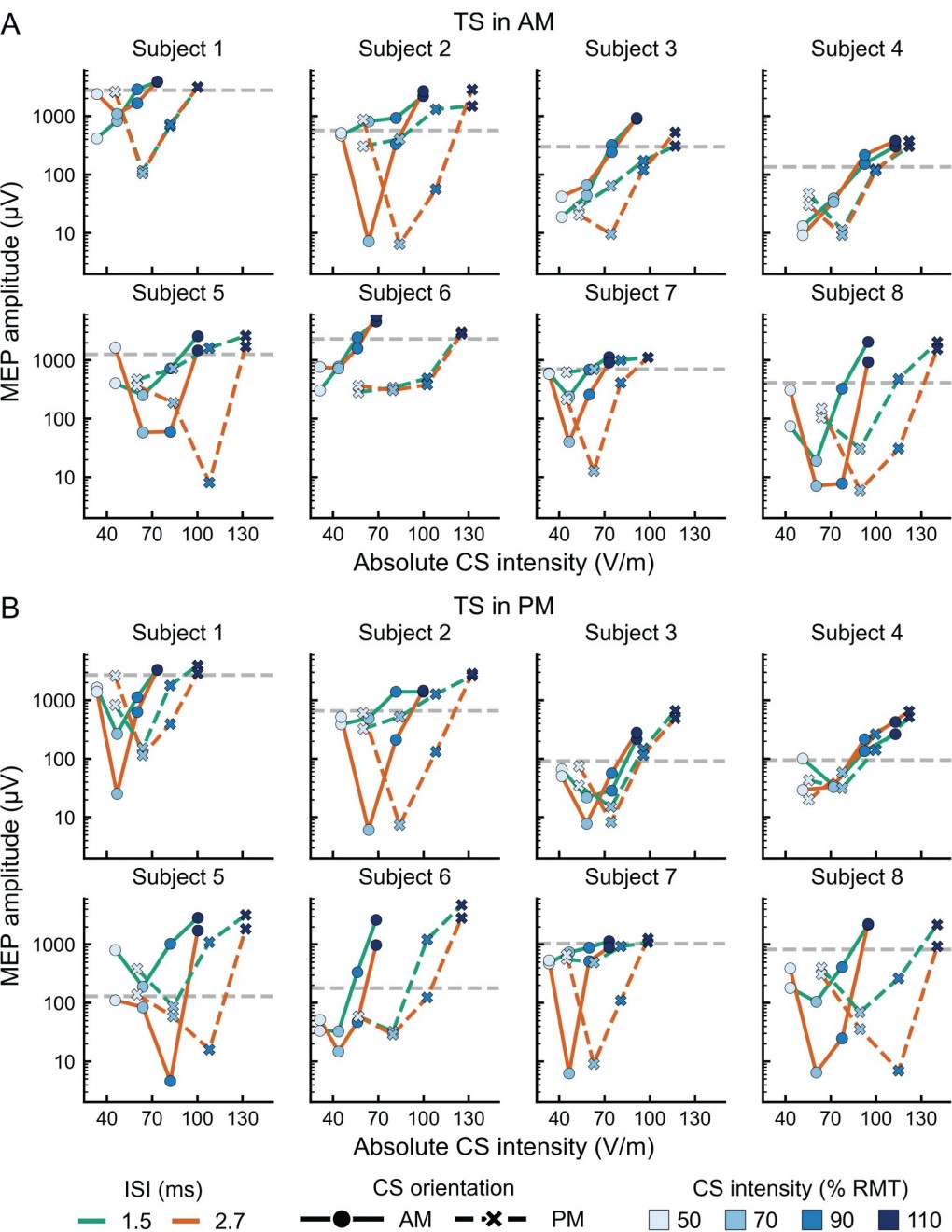

**Fig 3. MEP amplitude as a function of the absolute CS intensities in V/m.** TS in (A) AM and (B) PM orientations. The absolute CS intensities were computed as the RMT multiplied by the CS rising-phase duration (see Table 1). The absolute E-field intensities are subject-specific, thus, the data for each subject are presented in separate charts. The dashed, gray horizontal lines indicate the median amplitude of 20 single TS pulses.

curve for the CS in the PM orientation is shifted to the right by the orientation-specific difference compared to the SICI intensity curve for the CS in the AM orientation. The CS threshold for SICI is lower than the MEP threshold [8,10] but correlates with it [49]. It has been shown that SICI can be induced with a weaker stimulus intensity compared to the RMT and that the SICI threshold and different levels of inhibition along the SICI intensity curve correlate with

the individual RMT [49]. Therefore, the observed rightward shift of the SICI curve in the CS PM orientation by the MEP threshold difference further indicates that SICI is largely insensitive to the E-field orientation of the CS.

As a limitation to these lines of argument, it is important to note that even similar intensities of differently-oriented stimuli do not imply the same E-field distribution in the cortex due to the individual cortical anatomy [50,51]. This happens because the AM and PL stimulus orientations over the primary motor cortex produce the strongest E-field in the sulcus wall, while the PM and AL orientations produce a weaker E-field [52,53]. Thus, stimuli with the same intensity but different orientations have unequal potential to activate the pyramidal neurons; however, the influence of stimulus orientation on the inhibitory interneurons is not evident. Based on the above, we conclude that the finding of similar SICI for both AM and PM CS orientations in the previous studies [8,20] and our present results could be best explained by the insensitivity of the inhibitory interneurons to the E-field orientation.

Irrespective of the stimulus orientation, we observed stronger SICI at 2.7-ms ISI than at 1.5-ms ISI, with 70% and 90% RMT CS intensity. Stimuli at 2.7-ms ISI exhibit a lower facilitation potential for I-waves than do stimuli at 1.5 ms [10,48]; this can explain the stronger SICI effect observed in our experiment at the ISI of 2.7 ms.

## Effect of stimulus parameters on excitatory circuits

Stimulation with two consecutive supra-threshold pulses leads to similar MEP facilitation for all tested combinations of the CS and TS orientations. SICF possibly originates from the superposition of specific I-waves elicited by the CS and TS [17], potentiating the MEP amplitude. One likely explanation for the lack of orientation sensitivity of the SICF mechanism is based on the excitation of the neural populations generating I-waves. The neural populations could be equally excited in both CS orientations by the RMT adjustment. Both CS and TS were delivered with 110% RMT, each generating descending volleys that would result in an MEP. Thus, the lack of orientation sensitivity may support the view that the SICF mechanism originates from the superposition of the descending volleys rather than the intracortical balance of either inhibitory or excitatory inputs to the pyramidal neurons [4,15].

Additionally, the absence of difference between MEPs induced by paired-pulse stimuli at 1.5-ms and 2.7-ms ISIs can be partially explained by the supra-threshold intensity of both CS and TS. Therefore, the CS with supra-threshold intensity might produce a considerable change in the amplitude of the MEP. Moreover, Wagle-Shukla et al. demonstrated that increased intensity of CS led to a decrease of SICF effect and, thus, to the absence of a difference between MEP amplitudes at 1.5- and 2.7-ms ISIs [17]. In the same study, the authors suggested that CS with a high intensity independently activated most of the excitatory neurons and TS could activate only a limited number of these neurons, which did not lead to an increase of the SICF effect. We observed the same kind of saturation in our experiment; therefore, we speculate that the facilitation might have been induced by the summation of the cortico-spinal volleys from two consecutive supra-threshold stimuli.

Based on similar SICF responses for different stimulus orientations with 110% RMT for the CS and TS, our data suggest that the generation of SICF is not sensitive to the pulse orientations. In contrast, it was shown that another paired-pulse TMS facilitation phenomenon—ICF at longer latencies—is affected by the CS orientation when the CS intensity is fixed [8,20]. However, ICF has not been studied with the CS intensity adjusted to the orientation-specific RMT. Similarly, SICF has not been studied without the orientation-specific CS intensity for distinct E-field orientations. Thus, the genuine effect of the stimulus orientation on the SICF remains an open question.

### Methodological considerations

Our results were consistent across subjects and demonstrated SICI and SICF in most of the paired-pulse configurations. Nonetheless, interpretations should be considered carefully due to the limited sample size. The relatively small sample in the current study might lead to low statistical power, however the data were analyzed with a linear mixed-effects model which accounts for the inter-subject variability. We also anticipated large differences on the MEP amplitude between conditions (inhibition, facilitation, and stimuli orientations and intensities) providing a sufficient statistical power with limited sample size. More importantly, the primary goal of this study was to demonstrate the advantages of the normalizing CS and TS intensities based on the RMT. We highlight the importance of considering the stimulation intensity adjusted to the orientation-specific RMT when employing paired-pulse TMS with distinct orientations of the CS and TS. This intensity and orientation adjustment may help to disentangle the interaction between the inhibitory and excitatory mechanisms in the motor cortex also in other paired-pulse TMS paradigms, such as ICF, or long-interval intracortical inhibition.

## Conclusion

We applied paired-pulse mTMS to address the effect of the E-field orientation on SICI and SICF, highlighting the advantages of the multi-channel TMS technology in detailed investigation of the human cortex. In particular, the current study showed that both AM and PM orientations of the CS and TS can induce significant SICI and SICF in paired-pulse TMS protocols. SICI and SICF induced by stimulation in perpendicular orientations suggest that overlapping or strongly connected neuronal populations are activated. The minimal difference of SICI in all combinations of stimulus orientations indicates that inhibitory interneurons are largely insensitive to the E-field orientation, in contrast to the pyramidal cells. Finally, we revealed that SICI is insensitive to changes in the E-field orientation with orientation-specific stimulus intensity adjustment.

## Supporting information

**S1 Table. Multiple comparisons between interstimulus intervals (ISI).** The results are presented as follows, for a fixed conditioning stimulus orientation (CSO), test stimulus orientation (TSO), and conditioning stimulus intensity (CSI), we compute the motor evoked potential (MEP) amplitude ratio (MEP ratio) between the two tested ISIs. Each comparison has a standard error (SE), degrees of freedom (DoF), t-ratio, and p-value. Tested stimulus orientations were anteromedial (AM) and posteromedial (PM); CSI is given as a percentage of the orientation-specific resting motor threshold (RMT). *p*-values in bold are smaller than the threshold for statistical significance (0.05).
(DOCX)

**S2 Table. Multiple comparisons between conditioning stimulus orientations (CSO).** The results are presented as follows, for a fixed test stimulus orientation (TSO), conditioning stimulus intensity (CSI), and interstimulus interval (ISI), we compute the motor evoked potential (MEP) amplitude ratio (MEP ratio) between the two tested CSOs. Each comparison has a standard error (SE), degrees of freedom (DoF), *t*-ratio, and *p*-value. Tested stimulus orientations were anteromedial (AM) and posteromedial (PM); CSI is given as a percentage of the orientation-specific resting motor threshold (RMT). *p*-values in bold are smaller than the threshold for statistical significance (0.05).
(DOCX)

**S3 Table. Multiple comparisons between test stimulus orientations (TSO).** The results are presented as follows, for a fixed conditioning stimulus orientation (CSO), conditioning stimulus intensity (CSI), and interstimulus interval (ISI), we compute the motor evoked potential (MEP) amplitude ratio (MEP ratio) between the two tested TSOs. Each comparison has a standard error (SE), degrees of freedom (DoF), $t$-ratio, and $p$-value. Tested stimulus orientations were anteromedial (AM) and posteromedial (PM); CSI is given as a percentage of the orientation-specific resting motor threshold (RMT). $p$-values in bold are smaller than the threshold for statistical significance (0.05).
(DOCX)

**S4 Table. Multiple comparisons between condition stimulus intensities (CSI).** The results are presented as follows, for a fixed conditioning stimulus orientation (CSO), test stimulus orientation (TSO), and interstimulus interval (ISI), we compute the motor evoked potential (MEP) amplitude ratio (MEP ratio) between the tested CSI. Each comparison has a standard error (SE), degrees of freedom (DoF), $t$–ratio, and $p$-value. Tested stimulus orientations were anteromedial (AM) and posteromedial (PM); CSI is given as a percentage of the orientation-specific resting motor threshold (RMT). $p$-values in bold are smaller than the threshold for statistical significance (0.05).
(DOCX)

## Acknowledgments

The authors would like to thank Selja Vaalto, Olga Mikhailova and Denis Zakharov for their editorial support during the text preparation and productive discussions.

## Author Contributions

**Conceptualization:** Sergei Tugin, Victor H. Souza, Maria A. Nazarova, Pantelis Lioumis.

**Data curation:** Sergei Tugin, Victor H. Souza, Maria A. Nazarova, Pavel A. Novikov, Pantelis Lioumis.

**Formal analysis:** Sergei Tugin, Victor H. Souza, Pavel A. Novikov.

**Funding acquisition:** Risto J. Ilmoniemi.

**Investigation:** Sergei Tugin, Victor H. Souza, Maria A. Nazarova, Pantelis Lioumis.

**Methodology:** Sergei Tugin, Victor H. Souza, Maria A. Nazarova, Pavel A. Novikov, Pantelis Lioumis.

**Project administration:** Risto J. Ilmoniemi.

**Resources:** Risto J. Ilmoniemi.

**Software:** Sergei Tugin, Pavel A. Novikov, Aino E. Tervo, Jaakko O. Nieminen.

**Supervision:** Ulf Ziemann, Vadim V. Nikulin, Risto J. Ilmoniemi.

**Validation:** Sergei Tugin.

**Visualization:** Sergei Tugin, Victor H. Souza, Pavel A. Novikov.

**Writing – original draft:** Sergei Tugin.

**Writing – review & editing:** Sergei Tugin, Victor H. Souza, Maria A. Nazarova, Pavel A. Novikov, Aino E. Tervo, Jaakko O. Nieminen, Pantelis Lioumis, Ulf Ziemann, Vadim V. Nikulin, Risto J. Ilmoniemi.

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
