## [Decision Letter · Decision Letter 0]

7 Jul 2021

PONE-D-21-17054

Eﬀect of stimulus orientation and intensity on short-interval intracortical inhibition (SICI) and facilitation (SICF): a multi-channel transcranial magnetic stimulation study

PLOS ONE

Dear Dr. Tugin,

Thank you for submitting your manuscript to PLOS ONE. After careful consideration, we feel that it has merit but does not fully meet PLOS ONE’s publication criteria as it currently stands. Therefore, we invite you to submit a revised version of the manuscript that addresses the points raised during the review process.

We look forward to receiving your revised manuscript.

Kind regards,

Giuseppe Lanza, M.D., Ph.D.

Academic Editor

PLOS ONE

Journal Requirements:

2. Please change "female” or "male" to "woman” or "man" as appropriate, when used as a noun (see for instance https://apastyle.apa.org/style-grammar-guidelines/bias-free-language/gender).

“This project has received funding from the European Research Council (ERC) under the European Union’s Horizon 2020 research and innovation programme (grant agreement No 810377), the Jane and Aatos Erkko Foundation, the Academy of Finland (Decisions No. 294625, 306845, and 327326) and the Finnish Cultural Foundation. P. N. was funded by Aalto AScI Visiting Researcher Programme and by RFBR, project number 20-315-70048, M. N. was funded by RFBR, project number 20-315- 70048The authors would like to thank Selja Vaalto, Olga Mikhailova and Denis Zakharov for their editorial support during the text preparation and productive discussions.”

“This project has received funding from the European Research Council (ERC) under the European Union’s Horizon 2020 research and innovation programme (grant agreement No 810377), the Jane and Aatos Erkko Foundation, the Academy of Finland (Decisions No. 294625, 306845, and 327326) and the Finnish Cultural Foundation. Pavel A. Novikov was funded by Aalto AScI Visiting Researcher Programme and by RFBR, project number 20-315-70048. Maria A. Nazarova was funded by RFBR, project number 20-315-70048.”

Additional Editor Comments:

Dear Dr. Tugin,

as you can read below, both reviewers have appreciated your work, although they recommend some minor revisions. Therefore, I invite you to revise the paper accordingly in order to further proceed.

Best regards,

Giuseppe Lanza

Reviewers' comments:

Reviewer's Responses to Questions

**Comments to the Author**

1. Is the manuscript technically sound, and do the data support the conclusions?

Reviewer #1: Yes

Reviewer #2: Yes

2. Has the statistical analysis been performed appropriately and rigorously? 

Reviewer #1: Yes

Reviewer #2: Yes

3. Have the authors made all data underlying the findings in their manuscript fully available?

Reviewer #1: Yes

Reviewer #2: Yes

4. Is the manuscript presented in an intelligible fashion and written in standard English?

Reviewer #1: Yes

Reviewer #2: Yes

5. Review Comments to the Author

Reviewer #1: PONE-D-21-17054

The authors studied influences of the conditioning stimulus character (intensity and current direction) on the degree of SICI using their own newly developed multi-locus TMS transducer. Most of the results have confirmed the previously reported findings.

I have a few minor concerns.

The first paper focusing the direction of the conditioning stimulus has not been listed in the references. The same group reported a few papers concerning this issue.

The authors should list the following papers in References.

Hanajima R, Ugawa Y, Terao Y, Sakai K, Furubayashi T, Machii K, Kanazawa I.

Paired-pulse magnetic stimulation of the motor cortex: differences among I waves. J Physiol 509: 607-618, 1998

Hanajima R, Okabe S, Terao Y, Furubayashi T, Inomata-Terada S, Hamada M, Yugeta A, Ugawa Y. Difference in intracortical inhibition of the motor cortex between cortical myoclonus and focal hand dystonia. Clin Neurophysiol 119: 1400-1407, 2008

Hanajima R, Ugawa Y, Terao Y, Enomoto H, Shiio Y, Mochizuki H, Furubayashi T, Uesugi H, Iwata NK, Kanazawa I

Mechanisms of intracortical I-wave facilitation elicited by paired-pulse magnetic stimulation in humans. J Physiol 538: 253-261, 2002

The authors used two specific interstimulus intervals (1.5 and 2.7ms) for SICI. Why did they select these two intervals? Is it based on the authors’ previous experiments? The reason for the selection should be described in the method.

In the text, the authors mentioned that they performed four current directions of the conditioning stimulus in their previous experiments. However, the referred paper of this sentence is an abstract, and it has not reported this point precisely. In the present experiments, the authors used two directions angled 90 degrees. Why did they select these two directions in this paper? Why not used 180 degrees or 270 degrees. This reason should also be mentioned in the text.

Reviewer #2: The present study investigated how changes in the current direction of conditioning (CS) and test stimuli (TS) in a paired-pulse transcranial magnetic stimulation (TMS) paradigm affect the net outcome of the primary motor cortex. The main results were that posterior-medially (rotated 90 degree from the traditional posterior-anterior direction) oriented CS induced stronger short-interval intracortical inhibition (SICI) than the anterior-medially (tradition direction) oriented CS. Similar short-interval intracortical facilitation (SICF) was observed for CS delivered in both current directions. The study was well performed and the manuscript was well written. I only have a few minor comments. These are related to further discussion from a physiological viewpoint in addition to the most of current version of discussion from a technical viewpoint.

1, A specific hypothesis was raised at the end of the introduction that SICF but not SICI is sensitive to the CS current direction. The authors briefly mentioned that the hypothesis was driven from two previous studies (references 8 and 20, the recent one was a meeting proceeding). It would be helpful for the readers to better understand the background of the study if the authors describe more details in cortical physiology how they reached such a specific hypothesis.

2, A sample size with only eight subjects is relatively small for a TMS study. The statistical power for the study should be discussed.

3, It should be mentioned clearly if the focalities of two coils are same or not. The point may also be discussed.

4, The pulse durations for two stimuli were not mentioned clearly. The factor and how it interacts with the interstimulus intervals between two stimuli (if the duration changes with different interstimulus intervals) are important. These also need to be discussed.

5, SICF is elicited by two suprathreshold stimuli. It seemed that only TS induced MEP was accounted for the control MEP during data analysis. This should be discussed as both CS alone and TS alone produce a considerable size of MEP.

6. PLOS authors have the option to publish the peer review history of their article (what does this mean?). If published, this will include your full peer review and any attached files.

Reviewer #1: No

Reviewer #2: **Yes: **Zhen Ni, PhD, NINDS, NIH

---

## [Author Response · Author response to Decision Letter 0]

25 Aug 2021

Responses to editor

Comment 1: Please ensure that your manuscript meets PLOS ONE's style requirements, including those for file naming.

Response: We have modified the style of our manuscript and we are hoping that now it meets PLOS ONE's style requirements.

Comment 2: Please change "female” or "male" to "woman” or "man" as appropriate, when used as a noun (see for instance https://apastyle.apa.org/style-grammar-guidelines/bias-free-language/gender).

Response: We have now changed the “male” to “men”. You can find the updated sentence on Page 4, Section Methods, Subjects: 

“Eight healthy volunteers (aged 21–35 years, ﬁve men) with no contraindications to TMS [23,24] served as subjects after giving written informed consent.

Comment 3: Please remove any funding-related text from the manuscript and let us know how you would like to update your Funding Statement. Currently, your Funding Statement reads as follows:

“This project has received funding from the European Research Council (ERC) under the European Union’s Horizon 2020 research and innovation programme (grant agreement No 810377), the Jane and Aatos Erkko Foundation, the Academy of Finland (Decisions No. 294625, 306845, and 327326) and the Finnish Cultural Foundation. Pavel A. Novikov was funded by Aalto AScI Visiting Researcher Programme and by RFBR, project number 20-315-70048. Maria A. Nazarova was funded by RFBR, project number 20-315-70048.”

Response: We have removed the funding statement from the acknowledgment part. The updated version can be found on Page 14, Section Acknowledgements: 

“The authors would like to thank Selja Vaalto, Olga Mikhailova and Denis Zakharov for their editorial support during the text preparation and productive discussions.”

The version of the Funding Statement which you have provided is correct and we would not like to change it. We agreed that our Funding Statement reads as follows:

“This project has received funding from the European Research Council (ERC) under the European Union’s Horizon 2020 research and innovation programme (grant agreement No 810377), the Jane and Aatos Erkko Foundation, the Academy of Finland (Decisions No. 294625, 306845, and 327326) and the Finnish Cultural Foundation. Pavel A. Novikov was funded by Aalto AScI Visiting Researcher Programme and by RFBR, project number 20-315-70048. Maria A. Nazarova was funded by RFBR, project number 20-315-70048.”

Comment 4: Please review your reference list to ensure that it is complete and correct. If you have cited papers that have been retracted, please include the rationale for doing so in the manuscript text, or remove these references and replace them with relevant current references. Any changes to the reference list should be mentioned in the rebuttal letter that accompanies your revised manuscript. If you need to cite a retracted article, indicate the article’s retracted status in the References list and also include a citation and full reference for the retraction notice.

Response: We have revised the reference list and have organized it properly. We have added new articles according to recommendations of reviewer 1: Hanajima (1998, 2002, 2008). Additionally, our manuscripts that precede this work have been submitted for publication in peer-reviewed journals and the preprints are already available at bioRxiv:

Souza VH, Nieminen JO, Tugin S, Koponen LM, Baffa O, Ilmoniemi RJ. Probing the orientation specificity of excitatory and inhibitory circuitries in the primary motor cortex with multi-channel TMS. bioRxiv. 2021; https://www.biorxiv.org/content/early/2021/08/23/2021.08.20.457101

Souza VH, Nieminen JO, Tugin S, Koponen LM, Baffa O, Ilmoniemi RJ. Fine multi-coil electronic control of transcranial magnetic stimulation: effects of stimulus orientation and intensity. bioRxiv. 2021; https://www.biorxiv.org/content/early/2021/08/20/2021.08.20.457096

We added the citations to the preprints in the main text as they contain relevant information about the technical developments employed in this study.

 

Responses to reviewers

We thank the Reviewers for their constructive comments.

Reviewer #1: 

Minor comment 1: The first paper focusing the direction of the conditioning stimulus has not been listed in the references. The same group reported a few papers concerning this issue.

The authors should list the following papers in References.

Hanajima R, Ugawa Y, Terao Y, Sakai K, Furubayashi T, Machii K, Kanazawa I. Paired-pulse magnetic stimulation of the motor cortex: differences among I waves. J Physiol 509: 607-618, 1998

Hanajima R, Okabe S, Terao Y, Furubayashi T, Inomata-Terada S, Hamada M, Yugeta A, Ugawa Y. Difference in intracortical inhibition of the motor cortex between cortical myoclonus and focal hand dystonia. Clin Neurophysiol 119: 1400-1407, 2008

Hanajima R, Ugawa Y, Terao Y, Enomoto H, Shiio Y, Mochizuki H, Furubayashi T, Uesugi H, Iwata NK, Kanazawa I. Mechanisms of intracortical I-wave facilitation elicited by paired-pulse magnetic stimulation in humans. J Physiol 538: 253-261, 2002

Response: Thank you for indicating these issues. We agree that the papers by Hanajima are relevant and should be cited in our work. 

We have now added all the three suggested references:

Hanajima (2008) was added on Page 2, Section Introduction: 

“Paired-pulse transcranial magnetic stimulation (TMS) is widely used to assess the intracortical inhibitory and excitatory processes, offering a possibility to probe non-invasively the interactions of the cortical neuronal circuits in healthy and pathological conditions [4,5].

5. Hanajima R, Okabe S, Terao Y, Furubayashi T, Arai N, Inomata-Terada S, et al. Difference in intracortical inhibition of the motor cortex between cortical myoclonus and focal hand dystonia. Clin Neurophysiol. 2008;119(6):1400–7.”

Hanajima (1998, 2002) were added on Page 3, Section Introduction: 

“Due to the need for a millisecond-level ISI between the CS and TS, a standard TMS coil enabled the delivery of two consecutive pulses in same or oppositely directed orientations. The experiments with oppositely directed orientation of the E-field experiments play an important role for the identification of the interneurons for generation of I-wave and SICF [21,22]. However, the differently oriented current might have limited the minimal ISI to 3 ms [22,23] and not allowed to perform stimulation in other orientations. To overcome technical challenges of one orientation, … … …”

21. Hanajima R, Ugawa Y, Terao Y, Enomoto H, Shiio Y, Mochizuki H, Furubayashi T, Uesugi H, Iwata NK, Kanazawa I. Mechanisms of intracortical I-wave facilitation elicited with paired-pulse magnetic stimulation in humans. J Physiol. 2002 Jan 1;538(Pt 1):253-61. doi: 10.1113/jphysiol.2001.013094. PMID: 11773332; PMCID: PMC2290031.

22.Hanajima R, Ugawa Y, Terao Y, Sakai K, Furubayashi T, Machii K, Kanazawa I. Paired-pulse magnetic stimulation of the human motor cortex: differences among I waves. J Physiol. 1998 Jun 1;509 (Pt 2):607-18. doi: 10.1111/j.1469-7793.1998.607bn.x. PMID: 9575308; PMCID: PMC2230978.

Minor comment 2: The authors used two specific interstimulus intervals (1.5 and 2.7ms) for SICI. Why did they select these two intervals? Is it based on the authors’ previous experiments? The reason for the selection should be described in the method.

Response: The explanation was present on Page 6, Section: Method, SICI and SICF protocols: “ISI was either 1.5 or 2.7 ms to maximize the sensitivity to SICF [7,32,33], and both ISIs were effective for inducing SICI.”. We have now rephrased the explanation in order to make it more complete:

“ISI was either 1.5 or 2.7 ms to maximize the sensitivity to SICF [7,37,38]. Additionally, SICI induced with approximately 1 ms and 2,5-3 ms ISI has physiologically distinct phases and these intervals are widely applied to induce inhibition in paired-pulse protocols [11,39–41].”

Minor comment 3: In the text, the authors mentioned that they performed four current directions of the conditioning stimulus in their previous experiments. However, the referred paper of this sentence is an abstract, and it has not reported this point precisely. In the present experiments, the authors used two directions angled 90 degrees. Why did they select these two directions in this paper? Why not used 180 degrees or 270 degrees. This reason should also be mentioned in the text.

Response: We are thankful for this comment. Our previous manuscripts have been submitted for publication in a peer-reviewed journal and the preprints are already available at bioRxiv (Souza, 2021a; Souza, 2021b). We added the citations to the preprints in the main text. The results of our previous experiments indicate similar SICI effect when conditioning stimuli was oriented in either 90 degrees (posteromedial (PM)) or 270 degrees (anterolateral (AL)). Also, the 0 degrees (anteromedial (AM)) most common and ensured the strongest E-field at the sulcus wall (Janssen, 2015), and generates the highest MEP amplitudes. Thus, in this study our goal was to focus on the impact of stimuli intensities combining these two orientations rather than to cover all possible variations. Furthermore, the combination of AM and 180 degrees (posterolateral (PL)) orientations was not considered due to fact that this combination could be tested with the conventional coil (Delvendahl, 2013; Davila-Pérez, 2018; Hanajima, 1998). 

We are aware that the conventional TMS coil allows stimulation to be performed only with similar intensities of conditioning and test stimuli, regardless of their orientation. Additionally, performing stimulation through one coil limited the minimal interstimulus interval. Therefore, the paired-pulse experiments with AM and PL-oriented stimulus with normalized intensities might be scientifically novel and productive. We hope that this combination will be tested in future experiments. 

Since stimuli in the PM and AL orientations demonstrate similar amplitude of the MEP, we decided to only test one in this study. The PM orientation was chosen due to the fact that it is applied more often in paired-pulse protocols (Cirillo, 2016; 2018; Higashihara, 2020) compared to AL orientation. The further experiments should expand the orientations variety and build the database of the combination the role of orientation and intensity for the paired-pulses phenomena. 

Cirillo J, Byblow WD. Threshold tracking primary motor cortex inhibition: the influence of current direction. Eur J Neurosci. 2016 Oct;44(8):2614-2621. doi: 10.1111/ejn.13369. Epub 2016 Sep 1. PMID: 27529396.

Cirillo J, Semmler JG, Mooney RA, Byblow WD. Conventional or threshold-hunting TMS? A tale of two SICIs. Brain Stimul. 2018 Nov-Dec;11(6):1296-1305. doi: 10.1016/j.brs.2018.07.047. Epub 2018 Jul 18. PMID: 30033042. 

Davila-Pérez P, Jannati A, Fried PJ, Cudeiro Mazaira J, Pascual-Leone A. The Effects of Waveform and Current Direction on the Efficacy and Test-Retest Reliability of Transcranial Magnetic Stimulation. Neuroscience. 2018 Nov 21;393:97-109. doi: 10.1016/j.neuroscience.2018.09.044. Epub 2018 Oct 6. PMID: 30300705; PMCID: PMC6291364.

Delvendahl I, Lindemann H, Jung NH, Pechmann A, Siebner HR, Mall V. Influence of waveform and current direction on short-interval intracortical facilitation: a paired-pulse TMS study. Brain Stimul. 2014 Jan-Feb;7(1):49-58. doi: 10.1016/j.brs.2013.08.002. Epub 2013 Sep 10. PMID: 24075915.

Janssen, A.M., Oostendorp, T.F. & Stegeman, D.F. The coil orientation dependency of the electric field induced by TMS for M1 and other brain areas. J NeuroEngineering Rehabil 12, 47 (2015). https://doi.org/10.1186/s12984-015-0036-2

Hanajima R, Ugawa Y, Terao Y, Sakai K, Furubayashi T, Machii K, Kanazawa I. Paired-pulse magnetic stimulation of the human motor cortex: differences among I waves. J Physiol. 1998 Jun 1;509 (Pt 2):607-18. doi: 10.1111/j.1469-7793.1998.607bn.x. PMID: 9575308; PMCID: PMC2230978.

Higashihara M, Van den Bos MAJ, Menon P, Kiernan MC, Vucic S. Interneuronal networks mediate cortical inhibition and facilitation. Clin Neurophysiol. 2020 May;131(5):1000-1010. doi: 10.1016/j.clinph.2020.02.012. Epub 2020 Mar 3. PMID: 32193161.

Souza VH, Nieminen JO, Tugin S, Koponen LM, Baffa O, Ilmoniemi RJ. Probing the orientation specificity of excitatory and inhibitory circuitries in the primary motor cortex with multi-channel TMS. bioRxiv. 2021a; https://www.biorxiv.org/content/early/2021/08/23/2021.08.20.457101

Souza VH, Nieminen JO, Tugin S, Koponen LM, Baffa O, Ilmoniemi RJ. Fine multi-coil electronic control of transcranial magnetic stimulation: effects of stimulus orientation and intensity. bioRxiv. 2021b; https://www.biorxiv.org/content/early/2021/08/20/2021.08.20.457096

We have elaborated the reasons by which the current orientations were chosen. Please find the relevant corrections on Page 4, Section Methods, mTMS and stimulus definition:

“The AM and PM orientations were chosen due to the fact that they are widely applied in paired-pulse stimulation, albeit separately, i.e. in AM–AM or PM–PM combinations [34]. However, the combination of these orientations as a CS and TS was only tested twice before [8,20].” 

Reviewer #2 Zhen Ni: 

Minor comment 1: A specific hypothesis was raised at the end of the introduction that SICF but not SICI is sensitive to the CS current direction. The authors briefly mentioned that the hypothesis was driven from two previous studies (references 8 and 20, the recent one was a meeting proceeding). It would be helpful for the readers to better understand the background of the study if the authors describe more details in cortical physiology how they reached such a specific hypothesis.

Response: We realized that in our previous description of the hypothesis, the assertion that SICF but not SICI is sensitive to the CS current direction was not well-explained in the Introduction. Therefore, we explain the reasons for such conclusion in more detail on Pages 3–4, Section Introduction:

“Since SICI at 1–5 ms ISI required activation of the inhibitory interneurons, it can be assumed that this SICI is insensitive to the E-field orientation. Meanwhile, SICF required preactivation of the pyramidal neurons and therefore the E-field orientation is critical for it. Based on the above and also on previous studies [8,20], we hypothesized that a CS perpendicular to the TS would produce significant SICI and SICF. Additionally, changes in the CS orientation would critically impact SICF, while SICI phenomena would be more affected by changes in the CS intensity.”

Minor comment 2: A sample size with only eight subjects is relatively small for a TMS study. The statistical power for the study should be discussed

Response: We agree that only eight subjects is a relatively small sample. However, we anticipated a large difference between the conditions (inhibition, facilitation, stimuli and orientations and intensities) which would lead to sufficient statistical power even with a small sample size. For instance, we estimated that eight subjects provide an 80% statistical power when comparing two independent groups with anticipated mean MEP amplitudes 200 ± 50 µV and 100µV, with an alpha of 0.05. Most importantly, we analyzed our data using a linear mixed-effects model with subjects as a random effect. Such method provides a robust model that accounts for inter-subject variability. Additionally, we personalized the stimulation intensity and orientation following each subject’s motor responses. We have added a discussion about the statistical power on Page 13, Section Methodological considerations:

“The relatively small sample in the current study might lead to low statistical power, however the data were analyzed with a linear mixed-effects model which accounts for the inter-subject variability. We also anticipated large differences on the MEP amplitude between conditions (inhibition, facilitation, and stimuli orientations and intensities) providing a sufficient statistical power with limited sample size. More importantly, the primary goal of this study was to demonstrate the advantages of the normalizing CS and TS intensities based on the RMT.”

Minor comment 3: It should be mentioned clearly if the focalities of two coils are same or not. The point may also be discussed.

Response: We agreed that focality of the coils is a very important parameter. We have reported on this on Page 4, Section Methods, mTMS and stimulus definition: 

“The E-field focality was similar for all orientations.”

The experimental verification of coil focality is presented in the article “Fine multi-coil electronic control of transcranial magnetic stimulation: effects of stimulus orientation and intensity” where we introduced the multi-channel transducer which allowed to rotate the electric field. The corresponding article is available online via the link: https://www.biorxiv.org/content/early/2021/08/20/2021.08.20.457096

Specifically, we discussed this issue in the section Discussion, mTMS transducer:

“… This difference is due to the top coil’s 5-mm extra distance from the cortical surface, which leads to a weaker coil–cortex coupling (Koponen et al., 2017). The larger distance also leads to 3% (1–2 mm) poorer focality in perpendicular and parallel directions for the top compared to the bottom coil, a negligible difference for standard TMS applications. …

Koponen, L.M., Nieminen, J.O., Mutanen, T.P., Stenroos, M., Ilmoniemi, R.J., 2017. Coil optimisation for transcranial magnetic stimulation in realistic head geometry. Brain Stimul. 10, 795–805. https://doi.org/10.1016/j.brs.2017.04.001

We have added the reference to this statement on Page 4, Section Methods, mTMS and stimulus definition: 

“The E-field focality was similar for all orientations [28].”

[28] Souza VH, Nieminen JO, Tugin S, Koponen LM, Baffa O, Ilmoniemi RJ. Fine multi-coil electronic control of transcranial magnetic stimulation: effects of stimulus orientation and intensity. bioRxiv. 2021; https://www.biorxiv.org/content/early/2021/08/20/2021.08.20.457096

Minor comment 4: The pulse durations for two stimuli were not mentioned clearly. The factor and how it interacts with the interstimulus intervals between two stimuli (if the duration changes with different interstimulus intervals) are important. These also need to be discussed.

Response: We have presented three phases (rising, holding, and falling) duration for all the stimuli. The information about single pulse duration has been present on Page 4, Section Methods, mTMS and stimulus definition:

“Single pulses had a monophasic pulse waveform with 60.0-, 30.0-, and 43.2-µs rising, holding, and falling phases, respectively [30].”

Additionally, the exact times for all three phases were described for both CS and TS on Page 17, Section Supplementary Materials, Supplementary Table 1:

“Supplementary Table 1: TMS pulse durations of the conditioning (CS) and test stimuli (TS) in each CS intensity as a percentage of the resting motor threshold (RMT). The values represent the duration of three phases of a trapezoidal monophasic waveform: rising, maintaining, and falling.

CS intensities (% RMT) CS waveform durations rising, holding, falling (µs) TS waveform durations rising, holding, falling (µs)

50 24.8, 30.0, 19.3 72.3, 30.0, 51.0

70 37.1, 30.0, 27.9 73.2, 30.0, 52.0

90 51.2, 30.0, 37.4 76.7, 30.0, 53.7

110 70.5, 30.0, 49.8 82.8, 30.0, 57.5

”

We agree that this information is important, and we have moved this table from the supplementary section to Page 5, Section Methods, mTMS and stimulus definition.

The duration has not changed with different interstimulus intervals. However, the duration of rising and falling phases of the CS were increased respective to the increase of CS intensity. It is important to mention that the duration of the stimuli was increased, while the amplitude of the induced electrical field was fixed. Therefore, based on the previous studies (D'Ostilio, 2016; Nieminen, 2019; Peterchev, 2014) we assumed that intensity of the TMS can be equivalently controlled by adjusting either amplitude or duration of the induced electrical field.

The changes to the pulse duration do not affect the interstimulus Interval because the adjustments are on the order of tens of microseconds (min 74.1 µs; max 170.3), which is one order of magnitude smaller than the ISI. Furthermore, the neuronal membrane time constant is considerably higher than these small differences in pulse duration and should be similarly excited by both pulse waveforms.

D'Ostilio K, Goetz SM, Hannah R, Ciocca M, Chieffo R, Chen JA, Peterchev AV, Rothwell JC. Effect of coil orientation on strength-duration time constant and I-wave activation with controllable pulse parameter transcranial magnetic stimulation. Clin Neurophysiol. 2016 Jan;127(1):675-683. doi: 10.1016/j.clinph.2015.05.017. Epub 2015 May 30. PMID: 26077634; PMCID: PMC4727502.

Nieminen JO, Koponen LM, Mäkelä N, Souza VH, Stenroos M, Ilmoniemi RJ. Short-interval intracortical inhibition in human primary motor cortex: A multi-locus transcranial magnetic stimulation study. Neuroimage. 2019 Dec;203:116194. doi: 10.1016/j.neuroimage.2019.116194. Epub 2019 Sep 13. PMID: 31525495.

Peterchev AV, DʼOstilio K, Rothwell JC, Murphy DL. Controllable pulse parameter transcranial magnetic stimulator with enhanced circuit topology and pulse shaping. J Neural Eng. 2014 Oct;11(5):056023. doi: 10.1088/1741-2560/11/5/056023. Epub 2014 Sep 22. PMID: 25242286; PMCID: PMC4208275.

Minor comment 5: SICF is elicited by two suprathreshold stimuli. It seemed that only TS induced MEP was accounted for the control MEP during data analysis. This should be discussed as both CS alone and TS alone produce a considerable size of MEP.

Response: We agree that both CS and TS might induce MEP responses. To overcome this ambiguity, we added the following paragraph on Page 12, Section Discussion, Effect of stimulus parameters on excitatory circuits:

“Additionally, the absence of difference between MEPs induced by paired-pulse stimuli at 1.5-ms and 2.7-ms ISIs can be partially explained by the supra-threshold intensity of both CS and TS. Therefore, the CS with supra-threshold intensity might produce a considerable change in the amplitude of the MEP. Moreover, Wagle-Shukla et al. demonstrated that increased intensity of CS led to a decrease of SICF effect and, thus, to the absence of a difference between MEP amplitudes at 1.5- and 2.7-ms ISIs [17]. In the same study, the authors suggested that CS with a high intensity independently activated most of excitatory neurons and TS could activate only a limited number of these neurons, which did not lead to an increase of SICF effect. We observed the same kind of saturation in our experiment; therefore, we speculate that the facilitation might have been induced by the summation of the cortico-spinal volleys from two consecutive supra-threshold stimuli.”

---

## [Decision Letter · Decision Letter 1]

6 Sep 2021

Eﬀect of stimulus orientation and intensity on short-interval intracortical inhibition (SICI) and facilitation (SICF): A multi-channel transcranial magnetic stimulation study

PONE-D-21-17054R1

Dear Dr. Tugin,

We’re pleased to inform you that your manuscript has been judged scientifically suitable for publication and will be formally accepted for publication once it meets all outstanding technical requirements.

Kind regards,

Giuseppe Lanza, M.D., Ph.D.

Academic Editor

PLOS ONE

Additional Editor Comments (optional):

Reviewers' comments:

Reviewer's Responses to Questions

**Comments to the Author**

1. If the authors have adequately addressed your comments raised in a previous round of review and you feel that this manuscript is now acceptable for publication, you may indicate that here to bypass the “Comments to the Author” section, enter your conflict of interest statement in the “Confidential to Editor” section, and submit your "Accept" recommendation.

Reviewer #1: All comments have been addressed

Reviewer #2: All comments have been addressed

2. Is the manuscript technically sound, and do the data support the conclusions?

Reviewer #1: Yes

Reviewer #2: Yes

3. Has the statistical analysis been performed appropriately and rigorously? 

Reviewer #1: Yes

Reviewer #2: Yes

4. Have the authors made all data underlying the findings in their manuscript fully available?

Reviewer #1: Yes

Reviewer #2: Yes

5. Is the manuscript presented in an intelligible fashion and written in standard English?

Reviewer #1: Yes

Reviewer #2: Yes

6. Review Comments to the Author

Reviewer #1: The authors have responed to my previous comments well, and the contents of this paper attracts all reahders and is now acceptable for publication in Plos One.

Reviewer #2: I have no further comments.

I have no further comments.

I have no further comments.

I have no further comments.

7. PLOS authors have the option to publish the peer review history of their article (what does this mean?). If published, this will include your full peer review and any attached files.

Reviewer #1: No

Reviewer #2: **Yes: **Zhen NI

---

## [Editor Report · Acceptance letter]

13 Sep 2021

PONE-D-21-17054R1 

Eﬀect of stimulus orientation and intensity on short-interval intracortical inhibition (SICI) and facilitation (SICF): A multi-channel transcranial magnetic stimulation study 

Dear Dr. Tugin:

I'm pleased to inform you that your manuscript has been deemed suitable for publication in PLOS ONE. Congratulations! Your manuscript is now with our production department. 

Kind regards, 

on behalf of

Dr. Giuseppe Lanza 

Academic Editor

PLOS ONE